# Epidemiology and antimicrobial resistance of Enterobacteriaceae causing community-acquired urinary tract infections in Tétouan, Morocco (2022–2023)

Laila Farouk[1], Ayoub Ez-Zari[1]*, Lahcen Ouchari[2],
Zine El Abidine Bzazou EL Ouazzani[3], Zakaria Mennane[1], Noureddine El Mtili[1]

1 Laboratory of Biology and Health (UAE/U16FS), Department of Biology, Faculty of Sciences, Abdelmalek Essaqdi University, Tétouan, Morocco, 2 Molecular Biology and Functional Genomic Platform, National Center for Scientific and Technical Research (CNRST), Rabat, Morocco, 3 Grup de Biotecnologia Molecular i Industrial Department of Chemical Engineering Universitat Politècnica de Catalunya Terrassa, Terrassa, Spain

* ayoub.ezz1990@gmail.com

## Abstract

### Background

Urinary tract infections (UTIs) are among the most common community-acquired bacterial infections and Enterobacteriaceae are the leading etiological agents. Rising antimicrobial resistance (AMR) within this family poses major challenges for empirical treatment. This study aimed to describe the species distribution and antimicrobial resistance patterns of uropathogenic Enterobacteriaceae isolated from outpatients in Tétouan, Morocco.

### Methods

A cross-sectional descriptive study was conducted between April 2022 and December 2023 in three medical laboratories. Enterobacteriaceae isolated from urine cultures with significant bacteriuria were identified using the VITEK®2 system. Antimicrobial susceptibility testing was performed according to EUCAST 2021 guidelines. Extended-spectrum β-lactamase (ESBL) production was assessed using the Modified Double Disc Synergy Test (MDDST). Statistical analyses were performed using SPSS version 26, with significance set at $p < 0.05$.

### Results

A total of 422 Enterobacteriaceae isolates were obtained, predominantly from female patients (74.9%). *Escherichia coli* was the most frequent species (83.4%), followed by *Klebsiella pneumoniae* (9.2%). High resistance rates were observed for ampicillin (60.9%) and ticarcillin (56.2%), while resistance to imipenem (1.2%) and ertapenem

**Data availability statement:** All relevant data are within the manuscript and its Supporting Information files.

**Funding:** The author(s) received no specific funding for this work.

**Competing interests:** The authors have declared that no competing interests exist.

(0.9%) remained low. ESBL production was detected in 20 isolates (4.7%), with *E. coli* being the predominant ESBL-producing species (16/20; 80.0%), while *K. pneumoniae* exhibited a higher species-specific prevalence (4/39; 10.3%). Male patients exhibited significantly higher resistance to several β-lactams, while pediatric patients showed higher resistance to cephalosporins and aminoglycosides.

## Conclusion

Uropathogenic Enterobacteriaceae circulating in the community of Tétouan exhibit substantial resistance to commonly prescribed oral antibiotics, although carbapenems remain highly effective. The moderate prevalence of ESBL-producing strains highlights the need for reinforced antimicrobial stewardship and continuous regional surveillance to guide empirical treatment and limit the spread of resistant pathogens.

## Introduction

Urinary tract infections (UTIs) are among the most common bacterial infections globally, affecting both community and hospitalized populations and contributing to substantial morbidity and healthcare expenditure. Each year, an estimated 150 million individuals experience a UTI, underscoring the significant global burden of this condition [1–3]. UTIs occur when pathogenic microorganisms colonize and proliferate within the urinary tract, triggering an inflammatory response [2]. Clinically, UTIs are typically classified into uncomplicated infections—affecting otherwise healthy individuals with no urinary tract abnormalities—and complicated infections, which occur in the presence of structural or functional impairments, such as obstruction or renal dysfunction [4,5].

A wide range of microorganisms can cause UTIs, including Gram-negative and Gram-positive bacteria, fungi, and viruses. Nevertheless, members of the Enterobacteriaceae family—particularly *Escherichia coli (E. coli)*, *Klebsiella pneumonia (K. pneumonia)*, and *Proteus mirabilis (P. mirabilis)*—account for the majority of community- and hospital-acquired infections [6]. Effective management requires rapid identification of the causative pathogen and determination of its antimicrobial susceptibility profile. However, the inappropriate and excessive use of antibiotics has accelerated the emergence and dissemination of antimicrobial resistance (AMR), posing major challenges for empirical treatment and public health [7].

In response to the global rise in AMR, the World Health Organization (WHO) launched the Global Antimicrobial Resistance Surveillance System (GLASS) in 2015 to harmonize national reporting and enhance the comparability of resistance data across countries [8]. Morocco joined GLASS in 2018 and subsequently established a national coordination unit and technical committee to strengthen AMR surveillance [9]. Despite these national efforts, to our knowledge, published data on uropathogenic Enterobacteriaceae resistance in Morocco remain largely limited to a few urban centers, with northern regions, including Tétouan Province, particularly understudied.

The lack of updated regional evidence limits the ability of clinicians to tailor empirical therapy and restricts the development of context-appropriate stewardship strategies.

To our knowledge, no published study has specifically characterized the epidemiology and antimicrobial resistance of community-acquired uropathogenic Enterobacteriaceae in Tétouan Province. Generating region-specific data is essential to support national surveillance initiatives, guide empirical treatment choices, and contribute to AMR containment strategies.

Therefore, the present study aimed to characterize the species distribution and antimicrobial resistance profiles of community-acquired uropathogenic Enterobacteriaceae in Tétouan, Morocco, in order to provide region-specific evidence to inform clinical practice and support national AMR surveillance efforts.

## Materials and methods

### Ethical approval and participant consent

This study was conducted in accordance with ethical standards for research involving human participants. The study protocol was reviewed and approved by the Institutional Review Board (IRB) of the Hospital–University Ethics Committee of Tangier (CEHUT) (IRB No.: AC111JV/2-025), issued on 23 January 2022. The IRB waived the requirement for informed consent because this was a study based on routinely collected diagnostic samples and medical records. No written or verbal informed consent was obtained, as the ethics committee determined that consent was not required. All patient data were fully anonymized prior to access by the investigators and were used exclusively for research purposes. The study did not involve direct contact with patients.

### Study design and setting

A cross-sectional descriptive study was conducted between April 1, 2022, and December 31, 2023, in three clinical microbiology laboratories in Tétouan, Morocco, representing the three largest private medical laboratories located in the city center. Urine specimens were submitted for routine cyto-bacteriological examination of urine (ECBU) by attending physicians in outpatient settings, independently of this study. All consecutive positive cultures yielding Enterobacteriaceae from the three participating laboratories were eligible for inclusion, irrespective of patient demographics or clinical presentation. This study did not interfere with routine diagnostic procedures.

Significant bacteriuria was defined as ≥$10^5$ colony-forming units (CFU)/mL, in accordance with established diagnostic guidelines [10]. Cultures were performed on chromogenic agar and blood agar plates. Positive cultures yielding Enterobacteriaceae were subcultured to ensure purity and preserved at −80 °C in glycerol broth until further identification and antimicrobial susceptibility testing.

Tétouan Province, located in Morocco's northwestern Mediterranean zone, encompasses approximately 2,541 km² and had 573,784 inhabitants according to the 2019 provincial monograph [11].

### Inclusion and exclusion criteria

All Enterobacteriaceae isolates obtained from positive community-acquired urine cultures were included in the study. Duplicate isolates from the same patient with the same bacterial species were excluded to ensure each patient contributed only one isolate per episode of bacteriuria. Polymicrobial urine cultures and positive cultures for unidentified germs were also excluded.

### Identification and antimicrobial susceptibility testing

Identification of bacterial isolates was performed using the automated VITEK® 2 microbiology system (Biomerieux, France), employing the Gram-negative (GN) colorimetric reagent cards.

Antimicrobial susceptibility testing (AST) was conducted using the VITEK®2 GN AST-N233 card (Biomerieux, France). Results were interpreted according to the European Committee on Antimicrobial Susceptibility Testing (EUCAST) 2021 guidelines. Quality control was ensured using *Escherichia coli* ATCC 25922 as a reference strain. The VITEK®2 system categorized isolates as susceptible, intermediate, or resistant; intermediate isolates were grouped with the resistant category for analytical purposes.

The antimicrobial agents tested against the *Enterobacteriaceae* isolates included (with tested concentrations in µg/mL): ampicillin (4, 8, 32), amoxicillin–clavulanic acid (4/2, 8/4, 16/8), ticarcillin (16, 32, 64), cefoxitin (8, 16, 32), cefotaxime (1, 4, 16, 32), ceftriaxone (1, 4, 16, 32), ceftazidime (1, 2, 8, 32), imipenem (1, 2, 6, 12), ertapenem (0.5, 2, 4), amikacin (8, 16, 64), gentamicin (4, 16, 32), tobramycin (8, 16, 64), ciprofloxacin (0.5, 2, 4), and trimethoprim-sulfamethoxazole (1/19, 4/76, 16/304). Isolates exhibiting resistance to at least three distinct classes of antibiotics were classified as multidrug-resistant (MDR) [12].

### Detection of extended spectrum beta-lactamase production

Extended-spectrum β-lactamase (ESBL) production was detected using the Modified Double Disc Synergy Test (MDDST). A suspension of each test isolate, adjusted to $10^5$–$10^6$ CFU/mL, was inoculated onto Mueller–Hinton agar (MHA). A disc of amoxicillin–clavulanic acid (30/10 µg) was placed at the center of the plate. Discs of third-generation cephalosporins—ceftazidime (30 µg), cefotaxime (30 µg), ceftriaxone (30 µg)—and aztreonam (30 µg) were placed 30 mm (center-to-center) from the amoxicillin–clavulanic acid disc. Plates were incubated at 37 °C for 24 hours. ESBL production was confirmed by the presence of an enhanced inhibition zone toward the clavulanic acid disc, forming the characteristic "champagne cork" appearance [13].

### Data collection and statistical analysis

Data on patients' age, sex, urine culture results, etiological agents, and antimicrobial susceptibility profiles were extracted from clinical laboratory databases. Detailed strain information, including demographic and resistance characteristics, is provided in S1 Supporting Data in S1 Table. Data were organized using Microsoft Excel to evaluate the distribution of patients across the variables studied.

Univariate logistic regression analysis was performed to assess gender-related differences in UTI occurrence across age groups, with male sex as the reference category. Chi-square ($\chi^2$) tests were used to compare categorical variables, including resistance rates by gender and age group. Statistical significance was defined as a p-value $< 0.05$. All analyses were performed using IBM SPSS Statistics version 26 (IBM Corp., Chicago, IL, USA).

## Results

### Socio-demographic patient's characteristics

During the study period, a total of 422 clinical isolates of Enterobacteriaceae were collected. The majority were obtained from female patients (n = 316; 74.9%), resulting in a female-to-male ratio of 2.98. Participants' ages ranged from 12 days to 97 years, with a mean age of 42.5 ± 30.20 years. The mean age was higher among males (50.8 years) compared to females (39.7 years). The highest incidence of UTIs was observed in patients aged ≤17 years (n = 114; 27.0%). Univariate logistic regression analysis revealed significant gender disparities in UTI occurrence across several age groups. Females aged 18–24 years exhibited the highest odds of infection compared to males (OR = 13.94; 95% CI: 1.74–111.88; p = 0.013), followed by those aged 25–44 years (OR = 7.06; 95% CI: 2.56–19.49; p < 0.001) and 45–64 years (OR = 5.57; 95% CI: 1.88–16.56; p = 0.002). No statistically significant differences were observed in patients ≤17 years, 65–79 years, and ≥80 years (Table 1).

**Table 1. Sociodemographic characteristics and univariate logistic regression analysis of gender distribution across age groups among urinary tract infection patients in Tétouan, Morocco, 2022–2023 (N = 422).**

| Age Group (years) | Gender, n (%) | | Total, n (%) | OR [95% CI]* | P value |
|---|---|---|---|---|---|
| | Female | Male | | | |
| ≤ 17 | 84 (26.6) | 30 (28.0) | 114 (27.0) | 1.626 [0.79–3.32] | 0.183 |
| 18-24 | 24 (7.6) | 1 (0.9) | 25 (5.9) | 13.935 [1.74–111.88] | 0.013 |
| 25-44 | 73 (23.1) | 6 (5.6) | 79 (18.7) | 7.065 [2.56–19.49] | < 0.001 |
| 45-64 | 48 (15.2) | 5 (4.7) | 53 (12.5) | 5.574 [1.88–16.56] | 0.002 |
| 65-79 | 56 (17.7) | 46 (43.4) | 102 (24.2) | 0.692 [0.34–1.39] | 0.301 |
| ≥80 | 31 (9.8) | 18 (16.8) | 49 (11.6) | 1.626 [0.14–1.83] | 0.732 |
| Total | 316 (100) | 106 (100) | 422 (100) | | |

Age distribution by gender. Percentages in gender columns are calculated within each gender. Total column percentages are calculated over the entire sample (N = 422). * Odds ratios (OR) and 95% confidence intervals (CI) were obtained from univariate logistic regression with gender (female vs male) as the dependent variable.

## Enterobacteriaceae species distribution by age and gender

A total of 422 *Enterobacteriaceae* isolates were identified, with Escherichia coli being the predominant species (83.4%; n = 352), followed by *K. pneumoniae* (9.2%; n = 39) and *P. mirabilis* (3.1%; n = 13).

Gender distribution showed a predominance of female patients across most species, accounting for 77.3% (n = 272) of *E. coli*, 76.9% (n = 30) of *K. pneumoniae*, and 69.2% (n = 9) of *P. mirabilis* isolates. In contrast, *Klebsiella oxytoca* and other *Enterobacteriaceae* species were more frequently isolated from male patients, representing 75.0% (n = 6 out of 8) and 70.0% (n = 7 out of 10) of isolates, respectively.

Age distribution varied across species. Among *E. coli* isolates, the highest proportion was observed in patients aged ≤17 years (29.5%; n = 104), followed by those aged 65–79 years (22.4%; n = 79). *K. pneumoniae* was more frequently identified in older age groups, particularly among patients aged 65–79 years (28.2%; n = 11) and ≥80 years (17.9%; n = 7). Similarly, other *Enterobacteriaceae* species were predominantly observed in older patients, with 50.0% (n = 5) of isolates occurring in the 65–79 years group.

Overall, while *E. coli* predominated across all age groups, a relatively higher proportion of non-*E. coli Enterobacteriaceae* species was observed among older patients (Table 2).

## Antibiotic resistance profiles

Among the 422 clinical isolates analyzed, ampicillin (AMP) showed the highest resistance rate (61.1%; n = 258), followed by ticarcillin (56.2%; n = 237). In contrast, resistance to ertapenem (0.9%; n = 4) and imipenem (1.1%; n = 5) remained very low. Additionally, 70 isolates (16.8%) were multidrug-resistance (MDR).

**β-lactam resistance.** Among the penicillins, *E. cloacae* (n = 4) and *K. pneumoniae* (n = 39) exhibited the highest resistance to ampicillin (100% and 97.4%, respectively; n = 4/4 and n = 38/39) and amoxicillin–clavulanic acid (100% and 35.9%, respectively; n = 4/4 and n = 14/39). In contrast, *E. coli* (n = 352) and *P. mirabilis* (n = 13) showed lower resistance to ampicillin (55.1%; n = 194/352) and amoxicillin–clavulanic acid (15.4%; n = 2/13), respectively. For ticarcillin, *K. pneumoniae* (100%; n = 39/39) and *K. oxytoca* (100%; n = 4/4) were fully resistant, whereas *E. cloacae* displayed the lowest resistance (25%; n = 1/4).

Regarding cephalosporins, *K. pneumoniae* (n = 39) demonstrated the highest resistance to ceftriaxone (CRO), cefotaxime (CTX), and ceftazidime (CAZ), while *E. cloacae* (n = 4) showed the highest resistance to cefoxitin (FOX). *K. oxytoca* (n = 4) remained fully susceptible to all cephalosporins tested (0%; n = 0/4). Among carbapenems, the highest resistance was observed in *P. mirabilis*, with 4 of 13 isolates (30.8%) resistant to imipenem and 2 of 13 (15.4%) resistant

**Table 2. Distribution of Enterobacteriaceae species according to gender and age groups among patients with urinary tract infections in Tétouan, Morocco, 2022–2023 (N = 422).**

| Feature | Enterobacteriaceae species n (%) | | | | | |
|---|---|---|---|---|---|---|
| | E. coli | K. pneumoniae | P. mirabilis | K. oxytoca | Othersa | Total |
| **Gender** | | | | | | |
| Female | 272 (77.3) | 30 (76.9) | 9 (69.2) | 2 (25.0) | 3 (30.0) | 316 (74.9) |
| Male | 80 (22.7) | 9 (23.1) | 4 (30.8) | 6 (75.0) | 7 (70.0) | 106 (25.1) |
| **Age group (years)** | | | | | | |
| ≤ 17 | 104 (29.5) | 4 (10.3) | 3 (23.1) | 2 (25.0) | 1 (9.1) | 114 (27.0) |
| 18–24 | 18 (5.1) | 4 (10.3) | 1 (7.7) | 2 (25.0) | 0 (0.0) | 25 (5.9) |
| 25–44 | 67 (19.0) | 9 (23.1) | 3 (23.1) | 0 (0.0) | 0 (0.0) | 79 (18.7) |
| 45–64 | 46 (13.1) | 4 (10.3) | 1 (7.7) | 0 (0.0) | 2 (18.2) | 53 (12.5) |
| 65–79 | 79 (22.4) | 11 (28.2) | 3 (23.1) | 4 (50.0) | 5 (50.0) | 102 (24.2) |
| ≥80 | 38 (10.8) | 7 (17.9) | 2 (15.4) | 0 (0.0) | 2 (18.2) | 49 (11.6) |
| Total | 352 (100) | 39 (100) | 13 (100) | 8 (100) | 10 (100) | 422 (100) |

aOther species include: Enterobacter cloacae, Enterobacter aerogenes, Morganella morganii, Proteus penneri, Providencia rettgeri, Serratia marcescens, and Citrobacter koseri. Percentages are calculated within each Enterobacteriaceae species (column percentages). Percentages in the total column are calculated based on the overall sample size (N = 422).

to ertapenem. Given the small sample size of this species, these findings should be interpreted with caution. *K. oxytoca* (n = 4) and *E. cloacae* (n = 4) were fully susceptible to both agents (0%; n = 0/4).

**Aminoglycoside resistance.** *E. coli* (n = 352) exhibited the highest resistance to tobramycin (21.3%; n = 75/352), while *K. pneumoniae* (n = 39) showed the highest resistance to gentamicin (20.5%; n = 8/39).

*E. cloacae* (n = 4) displayed the highest resistance to amikacin (25%; n = 1/4).

**Quinolone and trimethoprim–sulfamethoxazole resistance.** *E. cloacae* (n = 4) exhibited the highest resistance to ciprofloxacin (25%; n = 1/4). *E. coli* (n = 352) demonstrated the highest resistance to trimethoprim–sulfamethoxazole (SXT) (36.6%; n = 129/352). *K. oxytoca* (n = 4) remained fully susceptible to both ciprofloxacin and SXT (0%; n = 0/4)

All antibiotic resistance profiles stratified by *Enterobacteriaceae* species and tested antimicrobial agents are summarized in Table 3.

## Prevalence, resistance profiles, and distribution of ESBL-producing isolates

Among the 422 Enterobacteriaceae isolates, 20 (4.7%) were confirmed as ESBL producers by MDDST. *E. coli* was the predominant ESBL-producing species, accounting for 16 of 20 ESBL isolates (80.0%), reflecting its overall abundance in the sample. When considering species-specific prevalence, 16 of 352 *E. coli* isolates were ESBL producers (4.6%), whereas 4 of 39 *K. pneumoniae* isolates were ESBL producers (10.3%), indicating a higher likelihood of ESBL production within *K. pneumoniae.*

ESBL-producing isolates showed complete resistance (100%) to tested penicillins (AMP, TIC) and third-generation cephalosporins (CRO, CAZ), while all remained susceptible to imipenem and ertapenem (Fig 1).

## Distribution of resistance patterns by gender and age

Male isolates exhibited significantly higher resistance rates to several β-lactam antibiotics compared with female isolates. Resistance to amoxicillin–clavulanic acid was observed in 37.7% of male isolates (n = 40/106) versus 24.4% of female isolates (n = 77/316; p = 0.009), while resistance to ampicillin was also more frequent among males (70.8%; n = 75/106 vs. 57.9%; n = 183/316; p = 0.026). Third-generation cephalosporin resistance, specifically to ceftriaxone and cefotaxime, was

**Table 3. Antibiotic resistance patterns of Enterobacteriaceae isolates among urinary tract infection patients in Tétouan, Morocco, 2022–2023 (N = 422).**

| *Enterobacteriaceae* Isolates | Antibiotics [a], n (%) | | | | | | | | | | | | | |
|---|---|---|---|---|---|---|---|---|---|---|---|---|---|---|
| | AMP | AMC | TIC | FOX | CRO | CTX | CAZ | IMP | ETP | AMK | GEN | TOB | CIP | SXT |
| *E. coli* (n = 352) | 194 (55.1) | 91 (25.9) | 182 (51.7) | 16 (4.6) | 22 (6.3) | 23 (6.5) | 27 (7.7) | 1 (0.3) | 1 (0.3) | 65 (18.5) | 69 (19.6) | 75 (21.3) | 81 (23.1) | 129 (36.6) |
| *K. pneumoniae* (n = 39) | 38 (97.4) | 14 (35.9) | 39 (100) | 4 (10.3) | 4 (10.3) | 3 (7.7) | 4 (10.3) | 0 | 1 (2.6) | 5 (12.8) | 8 (20.5) | 8 (20.5) | 6 (15.4) | 10 (25.6) |
| *P. mirabilis* (n = 13) | 8 (61.5) | 2 (15.4) | 5 (38.5) | 0 | 0 | 0 | 1 (7.7) | 4 (30.8) | 2 (15.4) | 1 (7.7) | 1 (7.7) | 1 (7.7) | 2 (15.4) | 2 (15.4) |
| *K. oxytoca* (n = 8) | 7 (87.5) | 2 (25) | 8 (100) | 0 | 0 | 0 | 0 | 0 | 0 | 0 | 0 | 1 (12.5) | 0 | 0 |
| *E. cloacae*[b] (n = 4) | 4 (100) | 4 (100) | 1 (25) | 3 (75) | 0 | 0 | 0 | 0 | 0 | 1 (25) | 0 | 0 | 1 (25) | 1 (25) |
| Others [c] (n = 6) | 6 (100) | 3 (50.0) | 2 (33.3) | 2 (33.3) | 0 | 0 | 1 (16.7) | 0 | 0 | 0 | 0 | 0 | 0 | 3 (50.0) |
| Total (N = 422) | 257 (60.9) | 116 (27.5) | 237 (56.2) | 25 (5.9) | 26 (6.2) | 26 (6.2) | 33 (7.8) | 5 (1.1) | 4 (0.9) | 72 (17.1) | 78 (18.5) | 85 (20.1) | 90 (21.3) | 145 (34.4) |

[a]Antibiotic abbreviations — AMP: Ampicillin; AMC: Amoxicillin–clavulanic acid; TIC: Ticarcillin; FOX: Cefoxitin; CRO: Ceftriaxone; CTX: Cefotaxime; CAZ: Ceftazidime; IMP: Imipenem; ETP: Ertapenem; AMK: Amikacin; GEN: Gentamicin; TOB: Tobramycin; CIP: Ciprofloxacin; SXT: Trimethoprim–sulfamethoxazole; [b] Enterobacter cloacae (n = 4) is presented as a separate row due to its clinically distinct resistance profile; all remaining minor species are grouped under 'Others' (n = 6); [c] Others: Enterobacter aerogenes, Morganella morganii, Proteus penneri, Providencia rettgeri, Serratia marcescens and Citrobacter koseri.

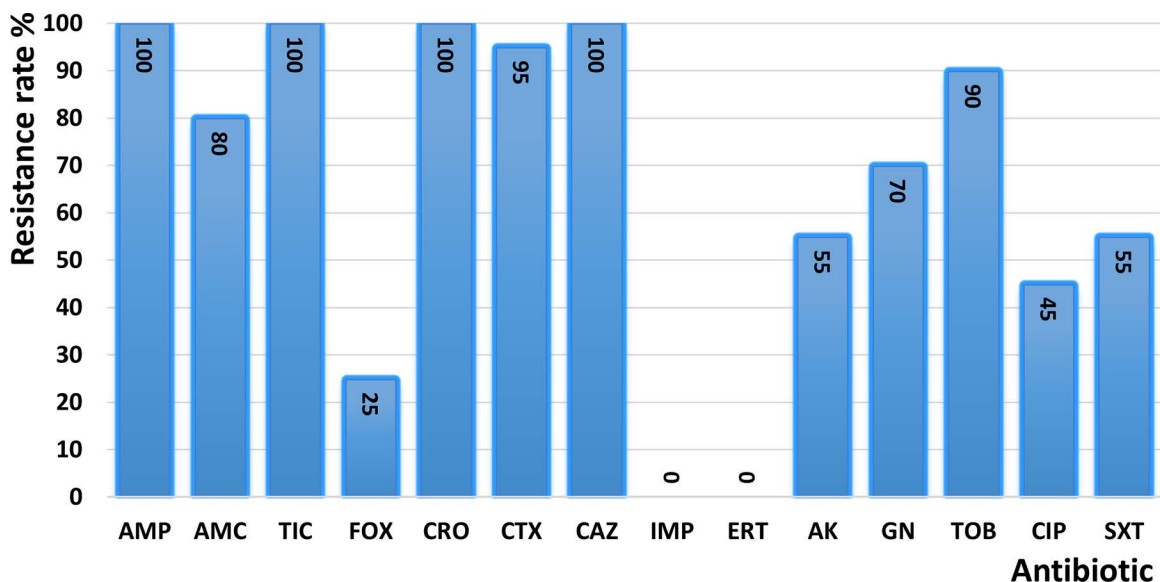

**Fig 1. Antibiotic resistance rate in uropathogenic extended-spectrum beta-lactamase producing isolates (n = 20) collected from outpatients in Tétouan, Morocco, 2022-2023.** The bar chart shows the percentage of resistance to tested antibiotics among extended-spectrum β-lactamase (ESBL)-producing uropathogenic Enterobacteriaceae isolates. Antibiotics tested include ampicillin (AMP), amoxicillin–clavulanic acid (AMC), ticarcillin (TIC), cefoxitin (FOX), ceftriaxone (CRO), cefotaxime (CTX), ceftazidime (CAZ), imipenem (IMP), ertapenem (ERT), amikacin (AK), gentamicin (GN), tobramycin (TOB), ciprofloxacin (CIP), and trimethoprim–sulfamethoxazole (SXT).

substantially greater in male isolates (13.2%; n = 14/106) than in female isolates (4.1%; n = 13/316; p = 0.002 for both). In contrast, resistance to imipenem and gentamicin did not differ significantly by gender.

Age-stratified analysis revealed differences in resistance patterns across pediatric, adult, and geriatric isolates. Cefoxitin resistance was highest among geriatric isolates (11.3%; n = 13/151, p = 0.034). Cefotaxime resistance was most prevalent in pediatric isolates (11.4%; n = 13/114; p = 0.033). Amikacin and gentamicin resistance were significantly higher in pediatric isolates (23.7%; n = 27/114, p = 0.018 and 25.4%; n = 29/114, p = 0.039, respectively). In contrast, ciprofloxacin resistance was highest in geriatric isolates (29.6%; n = 45/151) compared with adults (14.0%; n = 22/157) and pediatric isolates (21.1%; n = 24/114; p = 0.004).

For other antibiotics, although there were variations across age groups, these differences were not statistically significant (p > 0.05). Trimethoprim–sulfamethoxazole resistance remained consistently high across all age groups, while resistance to ticarcillin, tobramycin, and imipenem did not differ meaningfully by age.

ESBL prevalence was slightly higher among pediatric isolates (7.9%; n = 9/114) compared with adults (3.8%; n = 6/157) and geriatrics (3.3%; n = 5/151), but this difference was not statistically significant (p = 0.110). In contrast, ESBL occurrence was significantly greater in males (10.4%; n = 11/106) than in females (2.8%; n = 9/316; p = 0.001) (Table 4).

## Discussion

Urinary tract infections (UTIs) remain among the most frequent bacterial infections worldwide, with *Enterobacteriaceae* as the leading etiological agents [14]. Their ability to colonize the gastrointestinal tract and subsequently invade the urinary system underlies their high prevalence [2]. This study, the first from Tétouan, Morocco, provides updated insight into the epidemiology and antimicrobial resistance (AMR) of community-acquired *Enterobacteriaceae* UTIs.

**Table 4. Gender- and age-associated patterns of antimicrobial resistance among clinical Enterobacteriaceae isolates in Tétouan, Morocco, 2022–2023 (N = 422).**

| Resistance profile [a] | Gender, n (%) | | | Age group [b], n (%) | | | |
|---|---|---|---|---|---|---|---|
| | Male (n = 106) | Female (n = 316) | P value | Pediatric (n = 114) | Adult (n = 157) | Geriatric (n = 151) | P value |
| Amoxicillin–clavulanic acid | 40 (37.7) | 77 (24.4) | **0.009** | 37 (32.5) | 40 (25.5) | 40 (26.3) | 0.301 |
| Ampicillin | 75 (70.8) | 183 (57.9) | **0.026** | 69 (60.5) | 91 (58.3) | 97 (63.8) | 0.584 |
| Ceftriaxone (3GC) | 14 (13.2) | 13 (4.1) | **0.002** | 12 (10.5) | 7 (4.5) | 8 (5.3) | 0.435 |
| Cefoxitin | 8 (7.5) | 17 (5.4) | 0.281 | 4 (3.5) | 8 (5.1) | **13 (11.3)** | **0.034** |
| Cefotaxime | 14 (13.2) | 13 (4.1) | **0.002** | **13 (11.4)** | 6 (3.8) | 8 (5.3) | **0.033** |
| Ceftazidime | 14 (13.1) | 20 (6.3) | 0.260 | 13 (11.4) | 11 (7.1) | 10 (6.6) | 0.304 |
| Ticarcillin | 63 (58.9) | 175 (55.4) | 0.303 | 68 (59.6) | 85 (54.1) | 85 (55.9) | 0.636 |
| Imipenem | 1 (0.9) | 4 (1.3) | 0.627 | 2 (1.8) | 2 (1.3) | 1 (0.7) | 0.409 |
| Trimethoprim–sulfamethoxazole | 37 (34.6) | 109 (34.5) | 0.538 | 41 (36.0) | 48 (30.6) | 57 (37.5) | 0.362 |
| Ertapenem | 1 (0.9) | 3 (0.9) | 0.734 | 1 (0.9) | 1 (0.6) | 2 (1.3) | 0.684 |
| Amikacin | 18 (16.8) | 55 (17.4) | 0.510 | **27 (23.7)** | 27 (17.3) | 19 (12.5) | **0.018** |
| Gentamicin | 21 (19.6) | 58 (18.4) | 0.435 | **29 (25.4)** | 29 (18.5) | 21 (13.8) | **0.039** |
| Tobramycin | 25 (23.4) | 61 (19.3) | 0.221 | 28 (24.6) | 30 (19.1) | 28 (18.4) | 0.387 |
| Ciprofloxacin | 28 (26.2) | 63 (19.9) | 0.176 | **24 (21.1)** | 22 (14.0) | **45 (29.6)** | **0.004** |
| ESBL production [c] | 11 (10.4) | 9 (2.8) | **0.001** | 9 (7.9) | 6 (3.8) | 6 (3.9) | 0.110 |

Percentages are calculated using the total number in each subgroup (male, female, pediatric, adult, geriatric) as the denominator. Comparisons between groups were performed using the Chi-square test ($\chi^2$); P values < 0.05 were considered statistically significant. [a]Resistance profiles represent resistant isolates distributed by gender and age groups. [b] *Age group intervals: Pediatric 0–17 years, Adult 18–64 years, Geriatric ≥65 years.* [c]*ESBL production: isolates with extended-spectrum beta-lactamase production.*

Among 422 isolates, women represented the majority of cases (F/M ratio 2.98), consistent with national findings (F/M 1.1–2.17) [15–19] and international reports showing that 57–89% of UTIs occur in women [20–26]. This distribution is largely attributed to anatomical and physiological characteristics, as well as exposure factors such as pregnancy and sexual activity [14]. UTIs occurred across all ages, with a mean patient age of 42.5 years, aligning with national reports (34–47 years) [15,17–19]. Pediatric (≤17 years) and elderly (≥65 years) patients represented 27.0% and 35.8% of cases, respectively, highlighting their well-established vulnerability to infection. In children, this susceptibility is mainly ascribed to immune immaturity and anatomical factors [27], whereas in older adults, it is driven by immunosenescence and a higher burden of age-related comorbidities [28,29].

Stratified analysis revealed a marked overrepresentation of women aged 18–64 years, similar to observations in Morocco and elsewhere [20,22,30]. In contrast, no significant sex differences were found among children or the elderly, corroborating reports that gender disparities are less pronounced in these age groups [29]. Together, these findings illustrate the distinct epidemiological profile of UTIs across demographic strata.

*Escherichia coli* dominated the isolates (83.4%), consistent with national [15–19,30–32] and global trends [20,24,28,33–35]. Its predominance reflects its commensal presence in the gut and its arsenal of virulence factors, including adhesins, toxins, and biofilm-forming capacity [36,37]. *E. coli* was particularly common among women, children, and older adults, as widely reported [20,22,28,38]. *K. pneumoniae* was the second most frequent species (9.2%), in agreement with national data (10–31%) [16–19,30–32] and international findings identifying it as the main alternative pathogen after *E. coli* [20,22,24,25,28]. Other species such as *Enterobacter cloacae*, *Morganella morganii*, and *Citrobacter koseri* were each present at proportions below 1.5%, whereas *Klebsiella oxytoca* accounted for 1.9% of isolates and *Proteus mirabilis* accounted for a slightly higher proportion (3.1%), a pattern consistent with their typical involvement in complicated or nosocomial infections rather than community-acquired UTIs [20,24,39]. The higher proportion of non-*E. coli* *Enterobacteriaceae* among older patients may reflect increased comorbidities and greater cumulative healthcare exposure in this group [30].

High resistance rates were observed for ampicillin (60.9%) and ticarcillin (56.2%), in line with Moroccan reports [15,31,32,40] and lower than in several African settings where resistance often exceeds 70% [23,34]. These findings confirm the limited usefulness of older β-lactams for community-acquired UTIs. Resistance to ciprofloxacin (21.3%) and trimethoprim–sulfamethoxazole (34.4%) was also considerable. Comparable or higher levels have been reported in Morocco [15–18,31] and across Africa [20,23,34], underscoring persistent challenges for empirical therapy. Amoxicillin–clavulanic acid resistance (27.5%) was lower than in many Moroccan studies [16,17,31,32] but higher than in some European cohorts [20], suggesting it may retain partial value locally but requires close monitoring. Third-generation cephalosporins exhibited relatively low resistance rates (6.2–7.8%), consistent with Moroccan [17–19,30–32] and European data [20,21]. Overall carbapenem resistance remained rare (<2%), confirming their ongoing effectiveness [25,28,34], although emerging resistance in parts of Africa and the Middle East [23,28] reinforces the need for sustained surveillance.

ESBL-producing isolates accounted for 4.7% of cases, a prevalence lower than that reported from several Moroccan cities (≈9–10.5%) [15,31,40] but similar to that observed in the nearby city of Tangier (4.3%) [32]. Internationally, this prevalence is substantially lower than in Senegal (62%) [41] and Djibouti (46%) [25]. ESBL isolates displayed extensive resistance to β-lactams and elevated resistance to aminoglycosides, ciprofloxacin, and SXT, in keeping with co-resistance patterns described nationally [15,17,18]. Carbapenems in ESBL-producing isolates remained fully effective, underscoring their critical role as last-line agents.

*E. coli* showed high resistance to ampicillin (55.1%), ticarcillin (51.7%), and SXT (36.6%), with ciprofloxacin resistance at 23.1%. These rates mirror national data [15–17,32] and fall within the higher range of European studies [20,42]. Amoxicillin–clavulanic acid resistance (25.9%) was moderate and lower than in many Moroccan reports, suggesting possible retained efficacy. Aminoglycoside resistance (17–20%) exceeded some national averages, potentially reflecting local prescribing patterns. Third-generation cephalosporin resistance remained below 8%, consistent with the relatively low ESBL

prevalence. Carbapenem resistance was rare in *E. coli* (0.3%), although sporadic cases in neighboring regions [33,42] highlight the need for sustained monitoring. Third-generation cephalosporins remained largely effective against *E. coli* (<8% resistance), comparable to national (4–19%) [15,16,31,32,40] and European data (1.8–19.2%) [20,43]. These rates contrast sharply with high resistance reported in some African and Asian countries, where resistance may reach 100% [24,25,34,38,42]. This comparatively favorable profile supports their role as empirical options in Tétouan, in line with the relatively low ESBL prevalence [39,40].

*K. pneumoniae* exhibited near-universal resistance to ampicillin and ticarcillin, consistent with intrinsic penicillinase production [40]. Resistance to amoxicillin–clavulanic acid (35.9%) and aminoglycosides (12–20%) was moderate and aligned with national and international reports [15–18,28,32]. Third-generation cephalosporin resistance among *K. pneumoniae* (10.3%) remains concerning given this species' well-documented capacity to acquire and disseminate ESBL determinants [44]. Other Enterobacteriaceae demonstrated varied resistance profiles, though low sample numbers limit interpretation.

Notably, *P. mirabilis* isolates demonstrated higher resistance to carbapenems (including ertapenem and imipenem), an unexpected finding that warrants cautious interpretation in community settings. However, this observation should be interpreted with caution given the small sample size (n = 13). The reduced susceptibility may partly reflect intrinsic characteristics of *P. mirabilis*, such as decreased permeability associated with the absence of the OmpF outer membrane porin and its naturally lower susceptibility to certain carbapenems [45,46]. Nevertheless, molecular characterization was beyond the scope of this study, and further investigations with larger cohorts and carbapenemase screening are needed to better elucidate these findings and assess their potential epidemiological implications, particularly given that carbapenemase-mediated resistance may exhibit low phenotypic expression [33].

Demographic characteristics were associated with distinct antimicrobial resistance patterns in this cohort. Male patients exhibited higher resistance rates to amoxicillin–clavulanic acid and third-generation cephalosporins, as well as a greater proportion of ESBL-producing isolates, a pattern previously reported in community-acquired urinary tract infections [20,26]. Pediatric isolates showed increased resistance to aminoglycosides and cefotaxime, whereas isolates from elderly patients demonstrated higher resistance to ciprofloxacin and cefoxitin, findings often linked in the literature to age-specific antibiotic exposure and patterns of healthcare utilization [20,27,28,30]. These variations underscore the importance of considering demographic characteristics when interpreting local resistance trends. Nevertheless, they should be interpreted cautiously, as age and sex were analyzed as descriptive epidemiological factors rather than causal determinants of antimicrobial resistance.

The overall findings indicate a continued need for strengthened antimicrobial stewardship in Tétouan. Although resistance levels were lower than those reported in several Moroccan cities, upward trends remain concerning. Sustained regional surveillance, integrated within national AMR monitoring efforts, will be essential to track resistance evolution and to inform empirical therapy. Expanding access to routine susceptibility testing and improving diagnostic capacity may help reduce reliance on purely empirical treatment and facilitate earlier detection of ESBL-producing and other resistant strains across population groups.

This study has several limitations. It was conducted in a single province and included only outpatients, which may limit generalizability. Data on comorbidities and prior antibiotic use were unavailable, and ESBL detection relied on phenotypic methods without molecular confirmation. Future research should address these gaps through multicenter surveys, molecular characterization of resistance mechanisms, and exploration of alternative therapeutic strategies, including non-traditional agents.

## Conclusion

Community-acquired urinary tract infections in Tétouan were mainly caused by *E. coli*, with the highest burden observed among women and individuals at the extremes of age. The isolates showed substantial resistance to several first-line oral antibiotics, whereas the prevalence of ESBL-producing strains was moderate but clinically relevant.

These findings provide locally generated evidence on pathogen distribution and antimicrobial resistance patterns and underscore the need for ongoing regional surveillance to support empirical treatment decisions. Continued monitoring will be essential to detect emerging resistance trends and to inform antimicrobial stewardship efforts within the national context.

## Supporting information

**S1 Table. Individual-level antimicrobial susceptibility profiles of Enterobacteriaceae isolates from community-acquired urinary tract infections in Tétouan (2022–2023).** The table presents isolate-level data including bacterial species, patient sex, age, and antimicrobial susceptibility results. Antimicrobial susceptibility testing results are reported as S (susceptible), I (intermediate), or R (resistant) according to EUCAST criteria. Antibiotics tested include ampicillin (AMP), amoxicillin–clavulanic acid (AMC), ticarcillin (TIC), cefoxitin (FOX), ceftriaxone (CRO), cefotaxime (CTX), ceftazidime (CAZ), imipenem (IMP), ertapenem (ETP), amikacin (AMK), gentamicin (GEN), tobramycin (TOB), ciprofloxacin (CIP), and trimethoprim–sulfamethoxazole (SXT). Ages are reported in years; values "<1" indicate patients younger than one year. (XLSX)

## Acknowledgments

The authors express their sincere gratitude to the clinical private laboratories CHAMAL and ER-RAZI, located in Tétouan, for their valuable collaboration and support throughout this study.

## Author contributions

**Conceptualization:** Laila Farouk, Ayoub Ez-Zari, Noureddine El Mtili.

**Data curation:** Laila Farouk, Ayoub Ez-Zari, Zine El Abidine Bzazou EL Ouazzani.

**Formal analysis:** Laila Farouk, Zine El Abidine Bzazou EL Ouazzani.

**Investigation:** Laila Farouk, Ayoub Ez-Zari.

**Methodology:** Laila Farouk, Ayoub Ez-Zari, Zine El Abidine Bzazou EL Ouazzani, Zakaria Mennane, Noureddine El Mtili.

**Project administration:** Ayoub Ez-Zari, Lahcen Ouchari.

**Supervision:** Zakaria Mennane, Noureddine El Mtili.

**Validation:** Ayoub Ez-Zari, Lahcen Ouchari, Zine El Abidine Bzazou EL Ouazzani, Zakaria Mennane, Noureddine El Mtili.

**Visualization:** Zakaria Mennane.

**Writing – original draft:** Laila Farouk, Ayoub Ez-Zari.

**Writing – review & editing:** Ayoub Ez-Zari, Lahcen Ouchari, Zine El Abidine Bzazou EL Ouazzani.

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
