## [Decision Letter · Decision Letter 0]

19 Feb 2026

Dear Dr. EZ-ZARI,

Thank you for submitting your manuscript to PLOS ONE. After careful consideration, we feel that it has merit but does not fully meet PLOS ONE’s publication criteria as it currently stands. Therefore, we invite you to submit a revised version of the manuscript that addresses the points raised during the review process.

We look forward to receiving your revised manuscript.

Kind regards,

Dwij Raj Bhatta, PhD

Academic Editor

PLOS One

Journal Requirements:

3. We note that there is identifying data in the Supporting Information file “S1 Supporting information.xlsx”. Due to the inclusion of these potentially identifying data, we have removed this file from your file inventory. Prior to sharing human research participant data, authors should consult with an ethics committee to ensure data are shared in accordance with participant consent and all applicable local laws.

-Location data

Additional Editor Comments:

The manuscript be revised to make it suitable for publication. Please adress reviewers comments.

Reviewers' comments:

Reviewer's Responses to Questions

**Comments to the Author**

1. Is the manuscript technically sound, and do the data support the conclusions?

Reviewer #1: Partly

Reviewer #2: Yes

2. Has the statistical analysis been performed appropriately and rigorously?

Reviewer #1: No

Reviewer #2: Yes

3. Have the authors made all data underlying the findings in their manuscript fully available?

Reviewer #1: Yes

Reviewer #2: Yes

4. Is the manuscript presented in an intelligible fashion and written in standard English?

Reviewer #1: Yes

Reviewer #2: Yes

Reviewer #1: Farouk et al. described the antimicrobial resistance of bacteria of Enterobacteriaceae family causing urinary tract infections from the hospital/medical laboratories in Tetouan, Morocco. Only two demographic variables gender and age were used to describe antibiotic resistance in bacteria. Further, very limited epidemiological analysis has been done. Many results presented in tables and figures are redundant. The manuscript has limited scope and could be of interest of limited readers.

The following are the comments on the manuscript.

1. The short title looks like the key words without any correct meaning. Please correct it.

2. Although prospective descriptive study has been mentioned in the design, there is no analysis of data as per prospective design. The prospective design has a core aspect of the analysis to track how the measured variables change or evolve over the defined time frame. But there is no such analysis in the manuscript to monitor the trends.

3. ESBL E. coli were 72.7%, which was wrongly calculated. It should be calculated using ESBL E. coli as numerator and total E. coli tested as denominator.

4. The resistance trends according to age and gender of the patient have not been discussed. I think these variables although by chance found associated in the results but are not correct variables for comparison.

5. Page 3, line 70, please italicize Klebsiella pneumoniae.

6. Page 4, line 104, the prospective design is not appropriate word since there is no such analysis supporting prospective design.

7. Nitrofurantoin is the most commonly used antibiotic for the antibiotic susceptibility test of urine isolates. But it has not been found to be used in this study.

8. The distribution of bacterial species according to gender and age has been presented which is not relevant. Similar is the case for distribution of resistance patterns by gender and age. It cannot be explained that pediatric isolates were more likely to harbor resistance to 225 cephalosporins and certain aminoglycosides compared with adult and geriatric isolates.

9. Table 1 and 2 have presentation of same data. However, there is difference in some data between two tables such as isolates in age group 18-24, 45-64, 65-79 and 80 or more.

10. Table 3 and figure 1, and table 4 and figure 2, present the same data. Therefore, one is redundant.

11. As stated by authors, there are several limitations of the study. There are very limited data described based on very few variables.

Reviewer #2: The authors present a well-conducted prospective study addressing an important public health issue. The methodology is sound, and the findings provide valuable local data on antimicrobial resistance patterns, especially for the northern regions of Morocco. However, minor revisions are suggested to improve clarity, accuracy, and reader-friendliness.

Specific Comments:

1)In lines 61–62, the authors should consider citing more relevant and up-to-date references to strengthen the context.

2)Please ensure that abbreviations for binomial names, such as E. coli, are italicized throughout the manuscript. For examples, lines 171–172 require correction, also in line 70. Please check the entire text for consistency.

3)The methods are generally well described. However, the following points require attention. The phrase "in several medical laboratories" is vague. Please specify the number and type of laboratories (e.g., hospital-based, private, etc.) to enhance reproducibility. Moreover, specify whether consecutive or random sampling was used for isolate collection.

4)Please clarify that samples were only first episodes included, or were recurrent UTIs included? Were symptomatic and asymptomatic bacteriuria both included?

5)For consistency in data presentation, ensure all percentages throughout the manuscript (including the abstract) use the same number of decimal places. As an example, 4.97% could be rounded to 5.0% to match 74.7%, or both could be presented with two decimals (74.70% and 4.97%).

6)The presentation of Figures 1, 2, and 3 could be enhanced. I recommend improving the resolution, labels, and overall clarity to make the data more easily interpretable.

7)Lines 244–245 require revision to ensure accuracy. Please rephrase accordingly.

8)Line 266: Full spelling of Klebsiella pneumoniae is unnecessary as the abbreviation has already been introduced. Please revise and ensure scientific writing quality throughout.

9)The conclusion would be strengthened by briefly acknowledging study limitations (e.g., single region, short duration) to avoid overgeneralization.

.

Reviewer #1: **Yes:** Megha Raj BanjaraMegha Raj BanjaraMegha Raj BanjaraMegha Raj Banjara

Reviewer #2: No

---

## [Author Response · Author response to Decision Letter 1]

31 Mar 2026

Dear Editor-in-Chief,

We are pleased to submit the revised version of our manuscript entitled “Epidemiology and Antimicrobial Resistance of Enterobacteriaceae Causing Community-Acquired Urinary Tract Infections in Tétouan, Morocco (2022–2023)” (Manuscript ID: PONE-D-25-62063) for further consideration in PLOS ONE.

We sincerely thank you and the reviewers for your constructive and insightful comments. These suggestions have been invaluable in improving the clarity, rigor, and overall quality of the manuscript.

All reviewer and editor comments have been carefully addressed. A detailed, point-by-point response to each comment is provided, and all revisions have been tracked in the manuscript to ensure transparency.

In addition, the reference list has been thoroughly reviewed to ensure completeness and accuracy. Any retracted articles have been removed or replaced with appropriate, up-to-date references. Where relevant, retained retracted articles have been clearly identified as such in the reference list, with the corresponding retraction notices cited, in accordance with journal recommendations.

Note: During final verification of the dataset, we identified that one isolate initially included in the analysis had been misclassified as an Enterobacteriaceae species but was subsequently confirmed as Sphingomonas paucimobilis, which does not belong to the Enterobacteriaceae family. To preserve taxonomic accuracy, this isolate was removed from the dataset and all relevant calculations (including total N, percentages, and resistance rates) were recalculated using 422 confirmed Enterobacteriaceae isolates. The revised values differ only minimally from those originally reported and do not alter any of the study’s interpretations or conclusions. We have updated the manuscript accordingly and wish to emphasize that this correction improves the microbiological precision of the work without affecting its scientific integrity or the robustness of the findings.

We hope that the revised manuscript meets your expectations and we look forward to your favorable consideration.

Sincerely,

Ayoub Ez-Zari

Editor Requirements and comments:

1. Please ensure that your manuscript meets PLOS ONE's style requirements, including those for file naming. The PLOS ONE style templates.

Response: Thank you for your comment. We confirm that we have revised the manuscript to fully comply with PLOS ONE style requirements, including file naming conventions. The manuscript has been formatted according to the official PLOS ONE templates for the main body as well as the title, authors, and affiliations.

Response: We have revised the ethics statement to explicitly clarify the nature of the study, the full anonymization of data prior to access, and the IRB-approved waiver of informed consent, in accordance with the editor’s recommendations.

3. We note that there is identifying data in the Supporting Information file “S1 Supporting information.xlsx”. Due to the inclusion of these potentially identifying data, we have removed this file from your file inventory. Prior to sharing human research participant data, authors should consult with an ethics committee to ensure data are shared in accordance with participant consent and all applicable local laws.

Response: We thank the editor for highlighting this important data protection issue. The column entitled “Code” in the original S1 Supporting information.xlsx file was generated after anonymization of the dataset and was used solely for internal analytical purposes, namely to link biological samples with their corresponding phenotypic and demographic profiles. This code was not derived from, nor linked to, any patient identifiers, medical record numbers, or coordination systems.

Nevertheless, to fully comply with data-sharing and privacy requirements, we have entirely removed this column, along with any other variables that could be considered potentially identifying. The Supporting Information file has been revised accordingly and re-uploaded as a fully anonymized dataset whit no hiding data.

We confirm that the revised dataset is consistent with the ethical approval granted by the Institutional Review Board and that public data sharing does not compromise participant privacy or confidentiality.

Response: We have carefully reviewed the entire reference list to ensure its accuracy, completeness, and currency. All references have been checked for retraction status. No retracted articles are cited in the revised manuscript. All changes made to the reference list have been incorporated into the revised manuscript.

Review Comments to the Author:

Reviewer #1:

Farouk et al. described the antimicrobial resistance of bacteria of Enterobacteriaceae family causing urinary tract infections from the hospital/medical laboratories in Tetouan, Morocco. Only two demographic variables gender and age were used to describe antibiotic resistance in bacteria. Further, very limited epidemiological analysis has been done. Many results presented in tables and figures are redundant. The manuscript has limited scope and could be of interest of limited readers.

Response: We thank the reviewer for the constructive comments. Although only age and gender were analyzed, this reflects the nature of routinely available outpatient laboratory data and aligns with the descriptive objective of the study. The primary aim was to provide baseline, region-specific data on the antimicrobial resistance profiles of community-acquired Enterobacteriaceae in Tétouan rather than to conduct an exhaustive epidemiological risk factor analysis.

While the scope may appear limited, this study represents the first such investigation conducted at the provincial level in Tétouan and forms part of a broader series of studies on infectious diseases in this region. As such, it serves as a foundational reference for future, more detailed epidemiological and molecular investigations.

The central contribution of this work is to clarify the overall community resistance situation, where empirical antibiotic prescribing is common. These data provide locally generated evidence that can support updated empirical treatment recommendations and antimicrobial stewardship efforts.

All tables and figures were reviewed and revised to minimize redundancy and improve clarity.

We believe this study offers valuable regional surveillance data that contribute to the broader understanding of antimicrobial resistance in underreported North African community settings.

1. The short title looks like the key words without any correct meaning. Please correct it.

Response: We appreciate the reviewer’s suggestion regarding the short title. While the original short title was intended to be concise, we agree that it could be improved for clarity and readability. We propose the following alternative short titles: Antibiotic Resistance in Community UTI Enterobacteriaceae

2. Although prospective descriptive study has been mentioned in the design, there is no analysis of data as per prospective design. The prospective design has a core aspect of the analysis to track how the measured variables change or evolve over the defined time frame. But there is no such analysis in the manuscript to monitor the trends.

Response: We thank the reviewer for highlighting this point. Upon careful consideration, we acknowledge that our analysis does not track temporal changes in variables over the study period. Although the samples were collected between April 2022 and December 2023, the available data do not allow for time-resolved trend analysis. Therefore, to accurately reflect the study methodology and analysis, we have revised the manuscript to describe the study as a “cross-sectional descriptive study” rather than a prospective one. All instances of “prospective” in the manuscript, including the Abstract, Introduction, and Methods sections, have been updated accordingly. This modification ensures that the reported study design aligns with the analytical approach presented and maintains full transparency for the readers.

3. ESBL E. coli were 72.7%, which was wrongly calculated. It should be calculated using ESBL E. coli as numerator and total E. coli tested as denominator.

Response: We thank the reviewer for this important observation. We agree that reporting ESBL prevalence within each species provides a more structured and epidemiologically meaningful perspective. In our original manuscript, we reported the predominance of E. coli among all ESBL-producing isolates (16/20; 80%), reflecting its overall abundance in the sample. However, as the reviewer correctly points out, the species-specific prevalence (ESBL isolates of a species divided by total isolates of that species) gives a clearer understanding of the relative risk of ESBL production for each species.

In the revised manuscript, we have incorporated both approaches:

1. Composition of ESBL isolates: E. coli remains predominant among all ESBL isolates.

2. Species-specific prevalence: 4.6% of E. coli and 10.3% of K. pneumoniae isolates were ESBL producers, highlighting the higher likelihood of ESBL production in K. pneumoniae relative to its abundance.

We will also adopt the species-specific prevalence approach in all subsequent comparative analyses, as it provides a more accurate and meaningful assessment of resistance patterns.

4. The resistance trends according to age and gender of the patient have not been discussed. I think these variables although by chance found associated in the results but are not correct variables for comparison.

Response: We thank the reviewer for this valuable comment. Age and sex were included in our analysis as descriptive epidemiological variables, as commonly reported in antimicrobial resistance surveillance studies. In the revised manuscript, we have expanded the Discussion to contextualize the observed age- and sex-related resistance patterns with relevant literature and have explicitly clarified that these associations should be interpreted cautiously. We now emphasize that age and sex were analyzed as descriptive factors rather than causal determinants of antimicrobial resistance, thereby avoiding overinterpretation of these findings.

Discussion part: “Demographic characteristics were associated with distinct antimicrobial resistance patterns in this cohort. Male patients exhibited higher resistance rates to amoxicillin and third-generation cephalosporins, as well as a greater proportion of ESBL-producing isolates, a pattern previously reported in community-acquired urinary tract infections [20,26]. Pediatric isolates showed increased resistance to aminoglycosides and cefotaxime, whereas isolates from elderly patients demonstrated higher resistance to ciprofloxacin and cefoxitin, findings that have been commonly linked in the literature to age-specific antibiotic exposure and patterns of healthcare utilization [20,28,29,33]. These variations underscore the importance of considering demographic characteristics when interpreting local resistance trends; nevertheless, they should be interpreted cautiously, as age and sex were analyzed as descriptive epidemiological factors rather than causal determinants of antimicrobial resistance.”

5. Page 3, line 70, please italicize Klebsiella pneumoniae.

Response: done, and thank you for the remark

6. Page 4, line 104, the prospective design is not appropriate word since there is no such analysis supporting prospective design.

Response: We thank the reviewer for this important observation. Upon careful reconsideration, we agree that the term “prospective” was not appropriate, as the analysis did not include time-resolved or longitudinal assessments. Accordingly, we have revised the manuscript to describe the study as a descriptive cross-sectional study and have implemented all necessary modifications throughout the manuscript to ensure consistency with this study design.

7. Nitrofurantoin is the most commonly used antibiotic for the antibiotic susceptibility test of urine isolates. But it has not been found to be used in this study.

Response: We thank the reviewer for this pertinent observation. Nitrofurantoin is indeed commonly prescribed for urinary tract infections in Morocco and is included in routine urinary antibiograms. However, in our study, it was not displayed in the susceptibility results because our testing and reporting followed standard laboratory protocols, which restrict nitrofurantoin results to specific indications. Specifically: it is generally tested only for cystitis and primarily for E. coli isolates; it is not effective against all Enterobacteriaceae (e.g., Proteus spp., Pseudomonas spp.); and laboratory interpretative rules may limit its display when other antibiotics provide clearer guidance for therapy. Therefore, its absence in our reported panel reflects laboratory and reporting practices rather than an omission in study design.

8. The distribution of bacterial species according to gender and age has been presented which is not relevant. Similar is the case for distribution of resistance patterns by gender and age. It cannot be explained that pediatric isolates were more likely to harbor resistance to 225 cephalosporins and certain aminoglycosides compared with adult and geriatric isolates.

Response: We thank the reviewer for this observation. We acknowledge that age- and gender-specific differences in species distribution and resistance patterns cannot be interpreted as causal and should be considered descriptive only. While we did not emphasize these findings in the discussion, we reported them in the results to provide a complete epidemiological picture of our cohort. We have clarified in the manuscript that these associations are descriptive and should be interpreted cautiously, as they do not imply causation.

9. Table 1 and 2 have presentation of same data. However, there is difference in some data between two tables such as isolates in age group 18-24, 45-64, 65-79 and 80 or more.

Response: We thank the reviewer for pointing out the redundancy between Tables 1 and 2, as well as the discrepancies in some age groups. After careful review, we agree that presenting both tables separately was unnecessary. The discrepancies in the first table were due to the use of an unupdated version of the crude data.

To address this, we have merged the descriptive sociodemographic information and the univariate logistic regression ana

---

## [Editor Report · Decision Letter 1]

9 Apr 2026

Epidemiology and antimicrobial resistance of Enterobacteriaceae causing community-acquired urinary tract infections in Tétouan, Morocco (2022–2023)

PONE-D-25-62063R1

Dear Dr. Ez-Zari,

We’re pleased to inform you that your manuscript has been judged scientifically suitable for publication and will be formally accepted for publication once it meets all outstanding technical requirements.

Kind regards,

Dwij Raj Bhatta, PhD

Academic Editor

PLOS One
---

## [Editor Report · Acceptance letter]

PONE-D-25-62063R1

PLOS One

Dear Dr. Ez-Zari,

I'm pleased to inform you that your manuscript has been deemed suitable for publication in PLOS One. Congratulations! Your manuscript is now being handed over to our production team.

Kind regards,

on behalf of

Professor Dwij Raj Bhatta

Academic Editor

PLOS One